# A New Smoothed Seismicity Approach to Include Aftershocks and Foreshocks in Spatial Earthquake Forecasting: Application to the Global $M_w \geq 5.5$ Seismicity

**Matteo Taroni * and Aybige Akinci**

Istituto Nazionale di Geofisica e Vulcanologia, INGV Sez. Roma 1 Sismologia e Tettonofisica,
Via di Vigna Murata 605, 00143 Roma, Italy; aybige.akinci@ingv.it
* Correspondence: matteo.taroni@ingv.it

**Abstract:** Seismicity-based earthquake forecasting models have been primarily studied and developed over the past twenty years. These models mainly rely on seismicity catalogs as their data source and provide forecasts in time, space, and magnitude in a quantifiable manner. In this study, we presented a technique to better determine future earthquakes in space based on spatially smoothed seismicity. The improvement's main objective is to use foreshock and aftershock events together with their mainshocks. Time-independent earthquake forecast models are often developed using declustered catalogs, where smaller-magnitude events regarding their mainshocks are removed from the catalog. Declustered catalogs are required in the probabilistic seismic hazard analysis (PSHA) to hold the Poisson assumption that the events are independent in time and space. However, as highlighted and presented by many recent studies, removing such events from seismic catalogs may lead to underestimating seismicity rates and, consequently, the final seismic hazard in terms of ground shaking. Our study also demonstrated that considering the complete catalog may improve future earthquakes' spatial forecast. To do so, we adopted two different smoothed seismicity methods: (1) the fixed smoothing method, which uses spatially uniform smoothing parameters, and (2) the adaptive smoothing method, which relates an individual smoothing distance for each earthquake. The smoothed seismicity models are constructed by using the global earthquake catalog with $M_w \geq 5.5$ events. We reported progress on comparing smoothed seismicity models developed by calculating and evaluating the joint log-likelihoods. Our resulting forecast shows a significant information gain concerning both fixed and adaptive smoothing model forecasts. Our findings indicate that complete catalogs are a notable feature for increasing the spatial variation skill of seismicity forecasts.

**Keywords:** smoothed seismicity methods; global seismicity; foreshocks and aftershocks; earthquake forecasting model





## 1. Introduction

Building earthquake forecasting models is a fundamental step in any probabilistic seismic hazard analysis (PSHA). The spatial distribution of future seismicity is usually estimated using a seismicity catalog using two commonly adopted approaches called zonation [1,2] and smoothed seismicity [3,4]. In this work, we focus our attention on the smoothed seismicity approach. This approach uses statistical techniques to build a spatially gridded model using the epicenters of seismic events. One of the first examples of the smoothed seismicity model was developed by [3] and used the Gaussian isotropic spatial kernel to smooth the seismicity around epicenters. This model is based on only one parameter, i.e., the sigma of the Gaussian kernel: the larger the sigma, the larger the smoothing and vice versa. In the Frankel model, the sigma is fixed for any event, so it is called "fixed smoothed seismicity". Later, [4] developed a smoothed seismicity model that allows changing the sigma of the Gaussian kernel, and in general the size of any spatial

kernel function, according to the local density of earthquakes. The idea of this model is that where we have more events, we can use a smaller sigma to better define the seismic structures (i.e., the faults) that generate the seismicity. On the other hand, where we have fewer events, we can use a larger sigma to increase the coverage of the model in those lower seismogenic zones. In traditional PSHA, earthquakes are modeled using a Poisson process, where the occurrence of a future earthquake is independent of previous earthquakes from the same source [5]. The Poisson hypothesis holds for declustered catalogs. To include aftershocks and foreshocks within traditional PSHA, Ref. [6] presented an approach based on [7] theorem and its consequent generalization [8]. They demonstrated that the Poisson distribution could approximate the distribution of exceedances (also considering seismic sequences) in some specific conditions, e.g., for a probability of 10 percent or less of having an exceedance in 50 years (a typical value used for PSHA). Ref. [9] somewhat revised the initial [6] procedure. Rather than using their correction factor, Ref. [9] employed the b-value and the annual rate of the complete catalog as input for PSHA computations.

Both [6] and [9] suggest using a declustered seismic catalog only for the spatial estimation to avoid spatial bias introduced by the seismic sequence.

Therefore, a method that wants to introduce such sequences in the spatial estimation for PSHA needs a technique to downweigh the importance of aftershocks and foreshocks. Indeed, any seismic sequence should have the same importance in the spatial estimation of seismicity, independently from the number of events in the sequence (which can greatly vary between the sequences). The delcustering technique is the most dichotomous approach: it gives a weight equal to 1 to the mainshock and 0 to all other events in the sequence.

In their pioneering work, Ref. [10] developed a model to determine the spatial distribution of seismicity, including also the aftershocks and foreshocks in the seismic catalog. This approach uses a statistical model for the seismicity triggering, the ETAS model [11] and the stochastic declustering procedure [12] to assign each event the probability to be an independent event. In fact, in the ETAS model, events in the catalogs are distinguished as independent and dependent instead of mainshocks and aftershocks. The aftershocks of a seismic sequence, dependent on the sequence's mainshock, obtain a very low weight in this framework. Ref. [10] model consists of the multiplication of each spatial kernel for the probability to be independent of the associated earthquake. Therefore, in this framework, the spatial density distribution of a seismic sequence is mainly concentrated near the mainshock of the sequence (i.e., the independent event that generates all the dependent events of the sequence). Using this method, the fault that caused the seismic sequence is only partially reconstructed.

Our new, simple approach tries to solve that problem using a uniform weight for all the events of the same seismic sequence (i.e., $1/M$, where M is the number of events in the seismic sequence). In this manner, it is possible to describe the fault or the system of faults in a more coherent way, avoiding giving excessive weight to the mainshock of the sequence. Here, we use the global seismic catalog (CMT catalog), Ref. [13] to build four different spatial seismicity models, fixed and adaptive smoothed seismicity with and without our correction, to take into account the seismic sequences. Finally, we use the last ten years of the catalog to compare the performances of the models, using the spatial likelihoods of the models to measure their efficiency.

## 2. Dataset

We used the global centroid moment tensor (CMT) catalog containing 11,638 earthquakes with a depth $\leq$ 50 km recorded over the past almost 40 years between 1980 and 2019 [13,14]. We considered only events above the completeness magnitude as threshold $M_w$ = 5.5 [13,15]. The epicenter distribution of these events is shown in Figure 1. The current seismic sequences present in the seismic catalog have been detected by the [16] declustering algorithm, and the related parameters are provided and implemented in the ZMAP software [17]. Figure 2 shows the mainshocks (red dots) and foreshocks/aftershocks

(blue dots) in three zones in the world (Chile, Mexico, and Indonesia). Table 1 shows the number of events present in each subcatalog. We stress that the declustered catalog (i.e., the catalog containing only the mainshocks of the sequences) has 6440 events, about 45% less with respect to the complete catalog. This work aims to maintain as much data as possible and use all the available earthquakes in the catalog for the spatial distribution modeling. We underline that the use of a global catalog, instead of regional catalogs, has some drawbacks: a high threshold for the completeness magnitude (in our case $M_w$ 5.5), difficulty in recognizing volcanic events, and large uncertainties in hypocentral estimation. The main advantage is a large number of strong events, which can be collected in a few years.

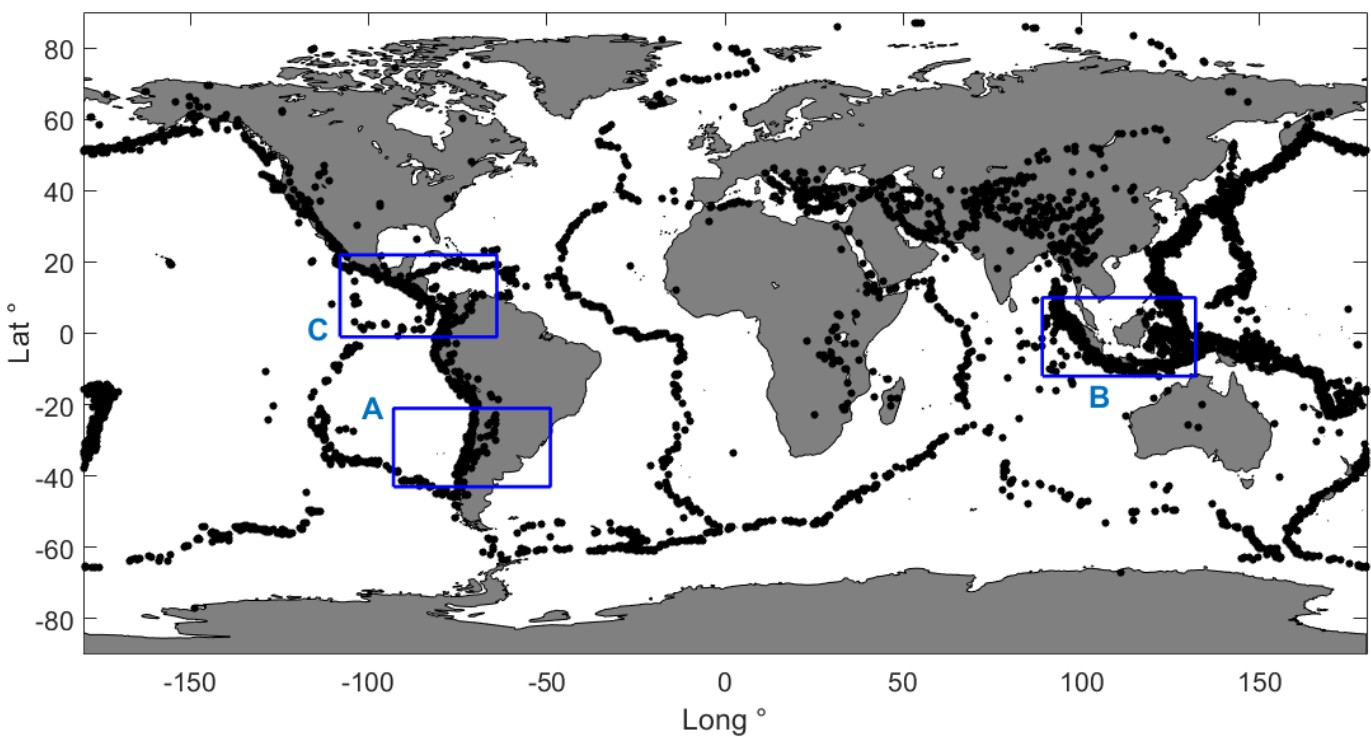

**Figure 1.** Location of earthquakes in the global centroid moment tensor (CMT) catalog with a depth ≤ 50 km recorded over the past almost 40 years between 1980 and 2019 [13,14]; blue letters indicate the zones of the zoom-in Figure 2.

**Table 1.** Number of events and time windows in the different catalogs from $M_w ≥ 5.5$.

| Catalog Type | Time Window | Number of Events |
|---|---|---|
| Complete | 1980–2019 | 11638 |
| Declustered | 1980–2019 | 6440 |
| Complete–Learning | 1980–2009 | 7977 |
| Declustered–Learning | 1980–2009 | 4718 |
| Complete–Testing | 2010–2019 | 3161 |

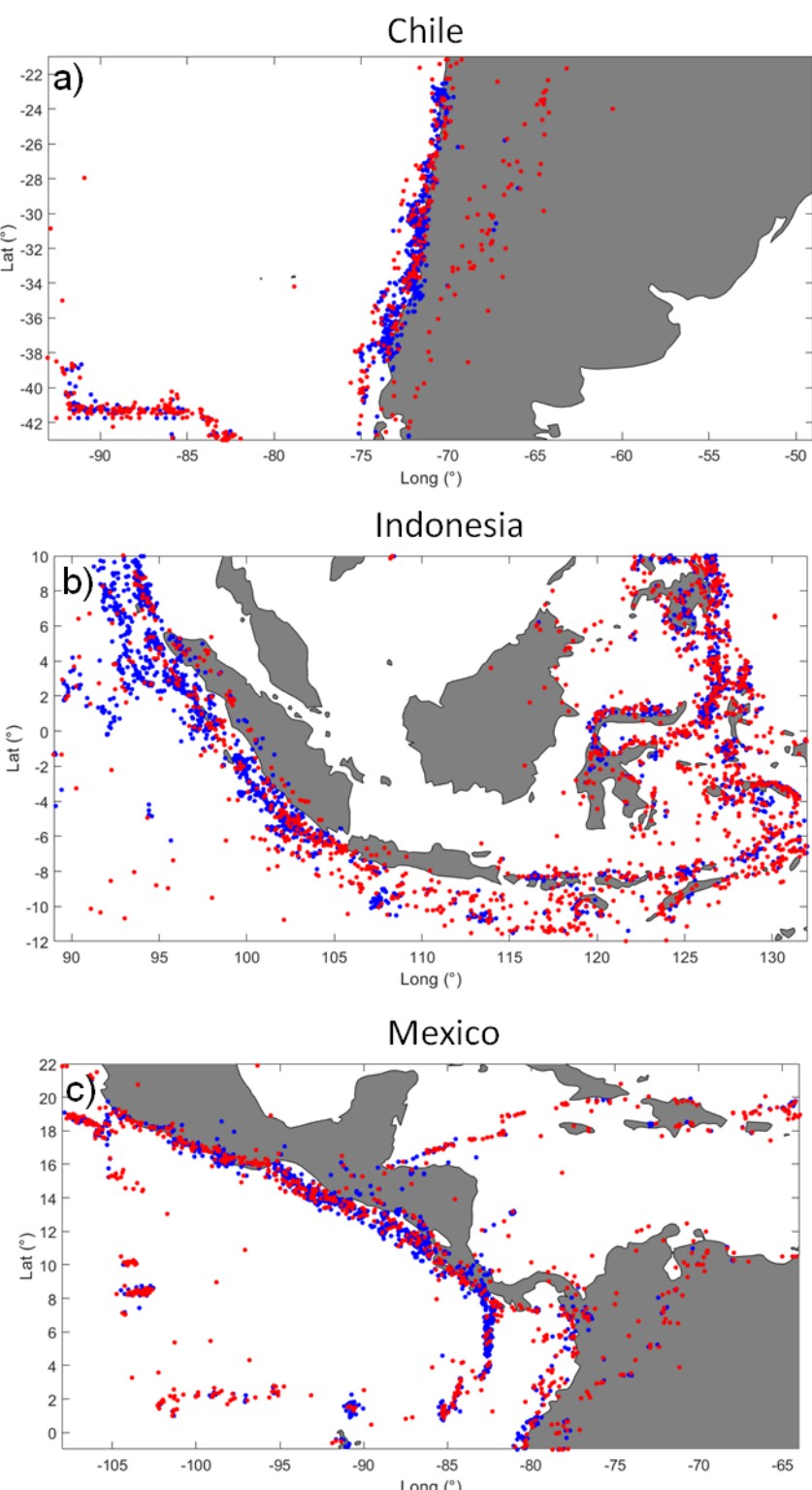

**Figure 2.** Location of earthquakes in the global centroid moment tensor (CMT) catalog with a depth ≤ 50 km recorded over the past almost 40 years between 1980 and 2019 [13,14]; (**a–c**) show the mainshocks (red dots) and foreshocks/aftershocks (blue dots) in some zones in the world: Indonesia, Mexico, and Chile.

## 3. A New Smoothed Seismicity Approach

Building a spatial grid is the first step to constructing a spatial smoothed seismicity model [3,4]. In this work, we used a global spatial regular grid, $0.5°$ by $0.5°$. Therefore, we need to compute the contribution to each event in the seismic catalog to the generic *i*-th spatial grid; the following equation describes that contribution:

$$f_i = \sum_{j=1}^{N} c K_{ij} A_i d_j \tag{1}$$

where $f_i$ represents the normalized total seismic rate for the *i*-th spatial grid, $N$ is the total number of events in the complete (i.e., not declustered) catalog, $c$ is the normalization factor $\left(c = \frac{1}{\sum f_i}\right)$, $K_{ij}$ is the kernel function that depends on the distance between the center of the *i-th* spatial cell and the epicentre of the *j*-th earthquake, $A_i$ is the area of the *i*-th spatial cell, and $d_j$ is the correction to take into account the foreshocks and aftershocks contribution to the spatial model.

The following Gaussian kernel function [3] is used:

$$K_{ij} = \frac{1}{2\pi\sigma^2} e^{-\frac{r_{ij}^2}{2\sigma^2}} \tag{2}$$

where $r_{ij}$ is the distance between the center of the *i*-th spatial cell and the epicentre of the *j*-th earthquake, and $\sigma$ is the free parameter of the model that rules the amplitude of the smoothing. However, we note that different kernel functions can also be employed in smoothing the epicenters from the earthquake catalog [4,18]. The smoothing distance, σ, involved in each earthquake may be defined differently in various smoothed seismicity models. For example, the fixed smoothed seismicity models practiced a single smoothing distance for all earthquakes. The adaptive smoothed seismicity models represent unique smoothing distances for each earthquake between an event and its *n*th closest neighbors (NN), resulting in spatially varying smoothing distances [4]. The distance becomes smaller in regions of high seismicity than in areas with sparse seismicity. It is one of the crucial parameters in the smoothed seismicity models both for the earthquake rates and the spatial variations of the earthquake activity rates in a region [19]. The correction parameter $d_j$ represents the innovative part of our method. It is defined as following $d_j = \frac{1}{S_j}$, where $S_j$ is the number of events in the seismic sequence and contains the *j*-th event. For example, if a seismic sequence contains ten events, one mainshock, and nine aftershocks, each event receives a weight, $= \frac{1}{10}$. Since the sum of all the weights is equal to one, the inclusion of aftershocks does not create a spatial bias in the model [6], and it leads to a better description of the fault that generated the sequence.

Using this simple correction may help better identify the active fault structures and their features in a region. Removing all the aftershocks and foreshocks [3,4], giving very high weight to the mainshocks only [10], may lead to an incomplete or biased view of the spatial distribution of future seismicity. Conversely, considering all the events in the sequence with a uniform weight, as in our method, increases the model's forecasting performance.

We underline that with Equation (1), we build normalized smoothed seismicity models, i.e., the sum of all the rates in the spatial cells are equal to 1. In this work, we do not face the problem of the total number of events and their magnitude frequency distribution, already treated in [9]. In that work, the seismicity rates are corrected by a proposed technique that allows counting all events in the complete seismic catalog by quickly adjusting the magnitude frequency relationships. Our method differs from theirs, since we only deal with the spatial distribution of the seismicity by using an equal weight for all the events of the corresponding seismic sequence and incorporating aftershocks to improve the spatial resolution of the model.

### 4. Likelihood Testing for Spatial Variation of Seismicity

To perform the maximum likelihood estimation of the parameter $\sigma$ (both for the fixed and adaptive smoothing approach) and to assess the performance of the model, we avoid considering the Poisson distribution of seismic events because this assumption is rarely satisfied by the seismic catalogs [20,21]. Since we are interested only in the spatial distribution of the events, and with Equation (1), we model the normalized spatial distribution of events, we defined the log-likelihood (*LL*) of the observations with:

$$LL(X|M) = \sum_{i=1}^{N} log(f_i) \tag{3}$$

where $X$ is the set of the $N$ observations (i.e., the epicenters of the events in the seismic catalog), $M$ is the spatial model, *log* is the natural logarithm, and $f_i$ is the seismic rate of the spatial cell where the $i$-th event is located. We note that this formulation differs from the spatial *LL* defined by [22] and has been commonly used in many seismic experiments [23], since the Poisson hypothesis has been abandoned in our study. The *LL* of Equation (3) may be ratified as the classical *LL* of a bivariate probability density function (represented by the model, *M*) in case we assume the independence between the observations in the set $X$. Additionally, in the case of nonindependent observations, the *LL* can be still used for scoring the models (some authors, in this case, called the function "pseudo-likelihood", [24].

To perform a pseudoprospective evaluation of the models first, we calculated the log-likelihood values by dividing the earthquake catalog into two parts: (1) the learning catalog, which contains the events recorded between 1980 and 2009 and is used to construct trial smoothed seismicity models, and (2) the testing catalog, which covers the last ten years of catalogs (2010–2019). The same *LL* of Equation (3) is also used to evaluate the performance of the models.

We applied the fixed and adaptive smoothing methods with and without our correction to include aftershocks and foreshocks for a total number of four different models. First, we used the learning catalog to compute the optimal smoothing parameters from the maximum-likelihood estimations (MLE), which strongly vary with smoothing distance (fixed smoothing) and neighbor number (adaptive smoothing). In the case of fixed smoothing, we used a vector of possible sigma (from 5 km to 200 km, with a spacing of 5 km), while for the adaptive smoothing, a set of possible neighbor numbers) are considered from 1 to 20, with a spacing of 1. The first part of the learning catalog (1980–1999) with a period of twenty years is utilized to build various smoothed seismicity models with different sigma and NN values. Finally, the nearest neighbor numbers and the correlation distances are calculated through maximum-likelihood optimization for the four smoothed seismicity models using the last ten years of the learning catalog (2000–2009). The results of these estimations are summarized in Table 2 and Figure 3.

We underline that these obtained MLE values are suitable only in the case of a global catalog: regional estimation of these parameters can lead to different MLE values (e.g., smaller sigma and larger NN).

**Table 2.** MLE of the parameters.

| Model | MLE |
|---|---|
| Fixed | Sigma = 135 |
| Adaptive | NN = 1 |
| Corrected Fixed | Sigma = 115 |
| Corrected Adaptive | NN = 1 |

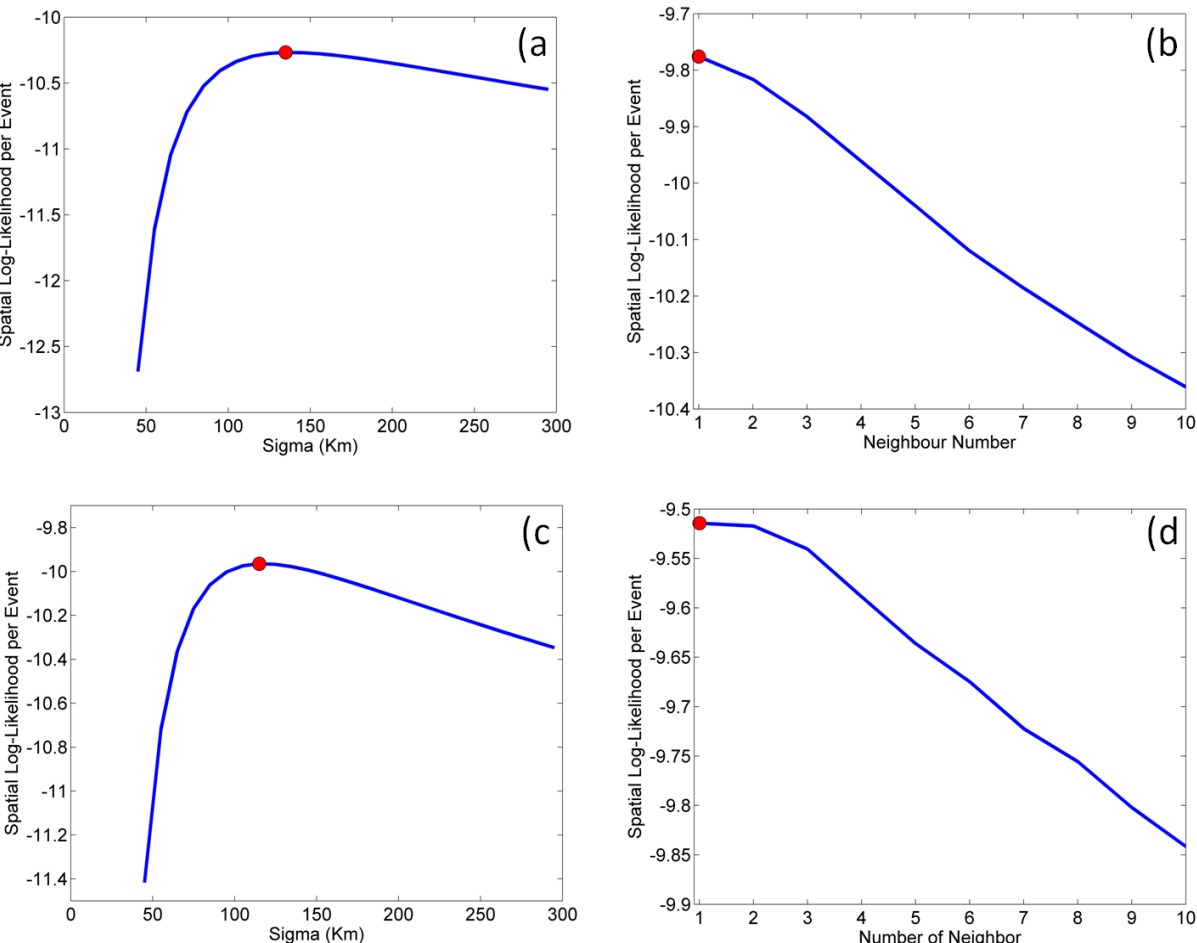

**Figure 3.** MLE for the parameters sigma and NN for the Fixed (**a**), Adaptive (**b**), Fixed$_{corrected}$ (**c**), and Adaptive$_{corrected}$ (**d**) models. Blue curves represent the log-likelihood functions, red dots the maximum of these functions.

## 5. Results

The final smoothed seismicity models are constructed using the entire learning catalog and the optimized correlation distances, previously obtained and given in Table 2. The models represent the bidimensional probability density function (PDF) of the seismicity (the sum of all the rates is 1). The corrected fixed smoothed seismicity model is calculated with a smoothing distance of 115 km, and it is 135 km in the case of the uncorrected model. Both adaptive smoothed seismicity models are determined using the nearest neighbor number equal to 1. These fixed and adaptive smoothed seismicity rate models are illustrated in Figure 4a,b (not corrected, hereafter fixed and adaptive) and Figure 4c,d (corrected, hereafter Fixed$_{corrected}$ and Adaptive$_{corrected}$).

To check if our corrected models perform better than those uncorrected smoothed seismicity models, we tested the Fixed$_{corrected}$ and Adaptive$_{corrected}$ models against the two standard fixed and adaptive smoothed seismicity models. Therefore, we performed a global pseudoprospective test, computing the *LL* (Equation (2)) of the four models using the ten-year testing catalog (2010–2019). Here, we outline that our testing catalog is entirely independent of the developed models. We preferred to endorse a similar computation procedure adopted in the real global prospective tests of the Collaboratory for the Study of Earthquake Predictability, CSEP, [23] and the global experiments [25,26]. We evaluated the performance of the models using two different magnitude thresholds, M$_w$ 5.5+ and M$_w$6.5+, to check the robustness of our models' forecasting locations and rates for future earthquakes. The results of these comparisons are presented in Tables 3 and 4 for the four developed models.

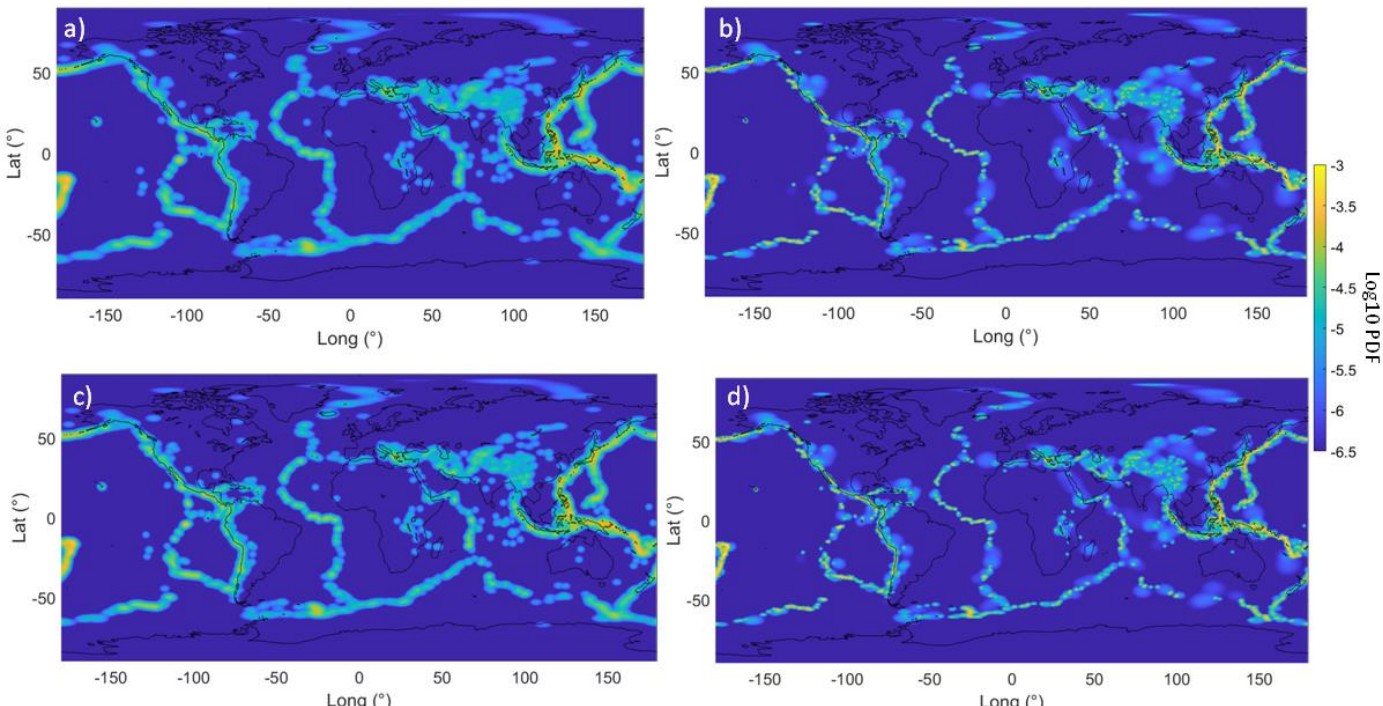

**Figure 4.** Spatially smoothed seismicity models using (**a**) 135 km smoothing distance from the fixed and (**b**) the nearest neighbor number NN = 1 from the adaptive smoothing seismicity approaches; spatially smoothed seismicity-corrected models using (**c**) 115 km smoothing distance from the fixed and (**d**) the nearest neighbor number NN = 1 from the adaptive smoothing seismicity approaches, employing the epicenters of the earthquakes for $M_w \geq 5.5$ in the global CMT catalog (normalized seismicity rates, i.e., PDF, are in log10 scale).

**Table 3.** Log-likelihood (*LL*) values for the smoothed seismicity models for testing catalog from magnitude $M_w$ 5.5 (3161 events).

| Model | Log-Likelihood (*LL*) |
|---|---|
| Corrected Adaptive | −29,632 |
| Adaptive | −29,639 |
| Corrected Fixed | −31,198 |
| Fixed | −31,297 |

**Table 4.** Log-likelihood (*LL*) values for the smoothed seismicity models for testing catalog from magnitude $M_w$ 6.5 (300 events).

| Model | Log-Likelihood (*LL*) |
|---|---|
| Corrected Adaptive | −2850 |
| Adaptive | −2857 |
| Corrected Fixed | −2931 |
| Fixed | −2949 |

For a correct interpretation of the models' *LL*, we recall that large *LL* values (i.e., the ones nearest to zero) indicate relatively good performances of the models, and small *LL* values (i.e., the ones further from zero) indicate relative bad performances of the models.

In general, our results show that the adaptive smoothed seismicity models (Adoptive and Adaptive$_{\text{corrected}}$) produce larger *LL* values and reveal better forecasting performance with respect to those from the fixed smoothed ones (Fixed and Fixed$_{\text{corrected}}$). The *LL*

values are −29,632 and −29,639 for the corrected and uncorrected adaptive smoothed models, while they are −31,297 and −31,198 in the case of the fixed corrected and corrected smoothed seismicity models, respectively (Table 3). The largest *LL* value calculated for the adopted smoothed seismicity models arises from the use of the correction parameter including the foreshocks and aftershocks in the global catalog. So, in general, including smaller earthquakes in the clusters increases the performance of the future $M_w \geq 5.5$ and $M_w \geq 6.5$ earthquake forecasting capability in the smoothed seismicity models.

To understand if this increase is rather significant, we interpreted the difference of the *LL* values for two models in terms of the Bayes factor [27], a common interpretation for pseudoprospective experiments [28–30]. According to [27] table, we obtained "very strong evidence" (difference in log-likelihood $\Delta LL > 5$) in favor of our proposed method, both for the fixed and adaptive approaches (Table 5). In Figure 5a–c, we also present the different maps calculated between the normalized seismicity rates (linear scale) of the adaptive and fixed corrected models (as $\text{Adaptive}_{corrected} - \text{Fixed}_{corrected}$), along with the events of the testing catalog, in the same zones of Figure 2: Indonesia (Figure 2a), Mexico (Figure 2b), and Chile (Figure 2c). Colors in light blue to red represent positive differences (i.e., the rate of the adaptive model is higher with respect to the fixed model), deep blue represents negative differences (i.e., the rate of the adaptive model is lower with respect to the fixed model), and blue represents no difference.

**Table 5.** Log-likelihood differences ($\Delta LL$) between the models.

| Models | Magnitude for the Comparison | Log-Likelihood Difference, $\Delta LL$ |
|---|---|---|
| Corrected Adaptive vs. Adaptive | 5.5+ | 7 |
| Corrected Adaptive vs. Adaptive | 6.5+ | 7 |
| Corrected Fixed vs. Fixed | 5.5+ | 99 |
| Corrected Fixed vs. Fixed | 6.5+ | 18 |
| Adaptive vs. Fixed | 5.5+ | 1658 |
| Adaptive vs. Fixed | 6.5+ | 92 |

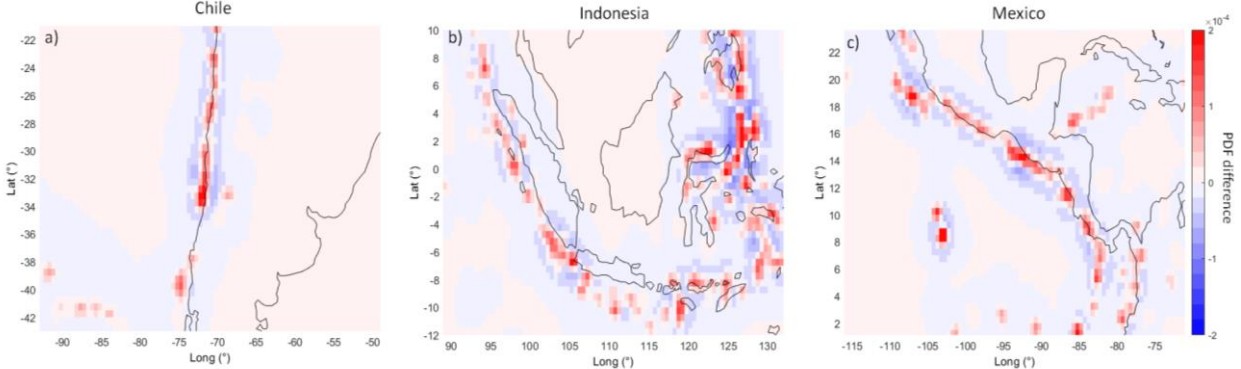

**Figure 5.** The difference between the normalized seismicity rates (linear scale) of the adaptive and fixed corrected models ($\text{Adaptive}_{corrected} - \text{Fixed}_{corrected}$) in some zones in the world: Chile (**a**), Indonesia (**b**), and Mexico (**c**).

## 6. Discussion

The comparison of the four global models, fixed and adaptive smoothed seismicity with and without our correction for the inclusion of aftershocks and foreshocks, clearly shows better performance of the models that use the correction. This positive result indicates that using all events of a seismic sequence instead of only the mainshock increases the forecasting capabilities of the smoothing seismicity models. Another very interesting result is the better performance of the adaptive approach concerning the fixed approach,

here demonstrated for a global catalog and two different magnitude thresholds, $M_w$ 5.5+ and 6.5+. Looking at the normalized seismicity rates in Figure 4a–d, it is possible to note the larger smoothing for the fixed models compared to the adaptive models in the zones where the seismicity is higher. The difference between the adaptive and fixed smoothing approaches is evidenced in Figure 5: the large smoothing for the fixed model leads to lower rates with respect to the adaptive model in the zones where the earthquake rate for the testing catalog is higher (pink and red colors in Figure 5). On the contrary, the rates of the fixed model are higher with respect to the adaptive model in the areas adjacent to the more seismic active zones (blue colors in Figure 5). Zones far from the main seismic regions (e.g., intraplate zones with very few earthquakes) have a very small difference between the fixed and adaptive seismicity rate models (light pink color in Figure 5).

The significantly better performances obtained by the adaptive smoothed approach (Table 5) confirm at a global scale the regional results obtained by [4] for California and [31] for Italy. Our method is more straightforward than that of [10], because it does not require a sophisticated stochastic declustering procedure [12]. Still, it only needs to identify the events in a seismic sequence, in this work made with the classical [16] declustering algorithm. Despite its simplicity, our method gives encouraging good results. A possible future work could be a comparison between our approach and the [10] approach.

Our method is based on the assumption of stationarity of the seismicity (usually accepted in long-term modeling); however, working in smaller time and spatial scales, some regions may exhibit different spatiotemporal variations, useful to forecast stronger seismic events [32,33]. Abandoning the stationarity assumption, smaller earthquakes can also be used to try to determine the current state of the seismic cycle [34] and then identify possible temporal variations in the long-term seismic rates.

## 7. Conclusions

The ten-year, global, pseudoprospective earthquake spatial forecasting experiment gives us two critical results:

(1) In general, the adaptive smoothing approach has better performance with respect to the fixed smoothing approach also for a global catalog with large events ($M_w \geq 5.5$ and $M_w \geq 6.5$);

(2) Using the simple correction described in this work, the inclusion of aftershocks and foreshocks leads to better spatial performances of the smoothed seismicity models.

A possible future improvement of our method is to include the events below the magnitude of completeness ($M_w < 5.5$) in the model to enhance and better describe the active fault structures and their segments.

**Author Contributions:** M.T. conceived the method; M.T. and A.A. defined the application; M.T. performed the data analysis and created the figures; M.T. and A.A. wrote the paper. Both authors have read and agreed to the published version of the manuscript.

**Funding:** This study was supported by Centro di Pericolosita' Sismica (CPS), Istituto Nazionale di Geofisica e Vulcanologia (INGV).

**Institutional Review Board Statement:** Not applicable.

**Informed Consent Statement:** Not applicable.

**Data Availability Statement:** Data and code used in this paper are available at: https://github.com/MatteoTaroniINGV/SmoothedSeismicity.

**Acknowledgments:** This study is under the framework of the Mappa di Pericolosita' Sismica, MPS16 Project supported by Centro di Pericolosita' Sismica (CPS), Instituto Nazionale di Geofisica e Vulcanologia (INGV).

**Conflicts of Interest:** The authors declare no conflict of interest.

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
