# Peer review of "A New Smoothed Seismicity Approach to Include Aftershocks and Foreshocks in Spatial Earthquake Forecasting: Application to the Global Mw ≥ 5.5 Seismicity"

_applsci, doi:10.3390/app112210899_

Round 1

Reviewer 1 Report

The article discusses various modifications of smoothing the distribution of sufficiently strong seismic events (with a magnitude of at least 5.5) on the Earth's surface using Gaussian kernel functions. The global seismic event catalog has been used since 1980. Kernel functions are considered isotropic. The result of the smoothing procedure is to obtain the values of the intensity of the seismic process at the nodes of the regular grid with a step of 0.5 degrees in latitude and longitude. The smoothing procedure is controlled by a parameter that has the meaning of the standard deviation of the Gaussian distribution. This parameter is chosen in two ways: either constant for all nodes of the regular grid, or adaptively, when the choice is made taking into account the spatial distribution of event epicenters in the vicinity of each node. For the method of choosing a constant value, an enumeration of trial values of the smoothing radius is performed from the minimum 5 km up to the maximum 200 km with a step of 5 km. The final choice of the smoothing parameter is carried out by solving the problem for the maximum likelihood function. In the process of smoothing, the aftershocks are not removed in order to more adequately assess the intensity of the seismic process for the tasks of engineering seismology, in which the main goal is to assess the shaking of the ground. The results of global seismicity smoothing are presented for the optimal constant value of the smoothing parameter (135 km) and for the adaptive method. As it would be expected, the map for the adaptive method shows the better spatial resolution.

Author Response

REVIEWER #1

COMMENT#1: The article discusses various modifications of smoothing the distribution of sufficiently strong seismic events (with a magnitude of at least 5.5) on the Earth's surface using Gaussian kernel functions. The global seismic event catalog has been used since 1980. Kernel functions are considered isotropic. The result of the smoothing procedure is to obtain the values of the intensity of the seismic process at the nodes of the regular grid with a step of 0.5 degrees in latitude and longitude. The smoothing procedure is controlled by a parameter that has the meaning of the standard deviation of the Gaussian distribution. This parameter is chosen in two ways: either constant for all nodes of the regular grid, or adaptively, when the choice is made taking into account the spatial distribution of event epicenters in the vicinity of each node. For the method of choosing a constant value, an enumeration of trial values of the smoothing radius is performed from the minimum 5 km up to the maximum 200 km with a step of 5 km. The final choice of the smoothing parameter is carried out by solving the problem for the maximum likelihood function. In the process of smoothing, the aftershocks are not removed in order to more adequately assess the intensity of the seismic process for the tasks of engineering seismology, in which the main goal is to assess the shaking of the ground. The results of global seismicity smoothing are presented for the optimal constant value of the smoothing parameter (135 km) and for the adaptive method. As it would be expected, the map for the adaptive method shows the better spatial resolution.

ANSWER#1:

We thank reviewer for the nice and the clear summary and positive answer.

Reviewer 2 Report

The paper contains the substantiation of a very interesting and correct, according to my opinion, idea of ​​how to take into account information on the position of the aftershock epicenters when assessing the spatial distribution of the probability of earthquakes, without overestimating local probabilities. On the one hand, if the distributions are constructed according to the declustered catalog, the detailing of fault systems on which earthquakes occur is lost. On the other hand, if the aftershocks are not removed in the estimates, too much weight will be attached to places with a large number of aftershocks. The authors found a nice compromise: to take into account the position of aftershocks, but when calculating the spatial probability density of earthquakes, each series of foreshock-main shock-aftershocks events is assigned a weight of 1 regardless of the number of events in the series. For declustering, the authors use the simplest Gardner-Knopoff method. It seems to me that this is quite justified. Moreover, it can be assumed that the result will depend very little on the choice of the declustering method. To prove the effectiveness of their approach, the authors conduct a pseudo-prospective test that gives good results.
The work deserves publication in the journal Applied Sciences as presented. It is only necessary to correct a small number of typos. For example, in lines 235 and 241 "Adopted" should be replaced with "Adaptive", or the text in lines 235-241 should be completely removed, since it repeats the data in Table 3.

Author Response

REVIEWER #2

COMMENT#1: The paper contains the substantiation of a very interesting and correct, according to my opinion, idea of ​​how to take into account information on the position of the aftershock epicenters when assessing the spatial distribution of the probability of earthquakes, without overestimating local probabilities. On the one hand, if the distributions are constructed according to the declustered catalog, the detailing of fault systems on which earthquakes occur is lost. On the other hand, if the aftershocks are not removed in the estimates, too much weight will be attached to places with a large number of aftershocks. The authors found a nice compromise: to take into account the position of aftershocks, but when calculating the spatial probability density of earthquakes, each series of foreshock-main shock-aftershocks events is assigned a weight of 1 regardless of the number of events in the series. For declustering, the authors use the simplest Gardner-Knopoff method. It seems to me that this is quite justified. Moreover, it can be assumed that the result will depend very little on the choice of the declustering method. To prove the effectiveness of their approach, the authors conduct a pseudo-prospective test that gives good results.
The work deserves publication in the journal Applied Sciences as presented.

ANSWER#1:

We thank reviewer for the nice and the clear summary and positive answer.

COMMENT#2: It is only necessary to correct a small number of typos. For example, in lines 235 and 241 "Adopted" should be replaced with "Adaptive", or the text in lines 235-241 should be completely removed, since it repeats the data in Table 3.

ANSWER#2:

We have corrected those suggested typos.

Reviewer 3 Report

This manuscript (ms) by Taroni and Akinci presents a new smoothed seismicity approach to include aftershocks and foreshocks in spatial earthquake forecasting. The application in made to the Global Mw≥5.5 seismicity and specific examples are discussed in detail for large regions around Indonesia, Mexico and Chile. The authors suggest an original and interesting improvement and the results convince the reader for the significance of the suggested new approach. The ms, however, although well written, needs improvements in the presentation. These improvements are:

1)In the Introduction, there is no comment on the very important and modern contribution of Earthquake Nowcasting which is closely related to the subject under study.

2)The letter size in all figures should be increased because it is very hard to read the labels. Moreover, in the multipaneled figures 2, 3, 4 and 5 panel labels (a), (b), (c) should be added.

3) The acronym NN seems to be multiply defined in the lines 130 and 183. This should be avoided. Moreover, for the readers’ better understanding an example of smoothing distances with actual values should be given either in the lines 122-134 or in the lines 181-184.  

4) The discussion in the lines 235 to 245 mixes the notion of “the smallest log-likelihood values” with the “better performance in forecasting future earthquakes” (simply -29632 > -31297). The authors should rephrase this paragraph to avoid confusion in the readers.

5)Last but not least, the present ms suffers from the following very serious shortcoming, which should be necessarily removed: Although the authors state in their abstract (line 12) that here they “present a technique to better determine future earthquakes in space”, surprisingly do not mentioned at all in their ms the recent work highlighted by PNAS, entitled “Spatiotemporal variations of seismicity before major earthquakes in the Japanese area and their relation with the epicentral locations” [PNAS 112, 986-989 (2015)], which presents a modern approach to estimate the epicentral location of forthcoming earthquakes. Thus, I would suggest to the authors to add the following (probably as a separate paragraph in their “Discussion” Section 6):

“A different approach based on natural time analysis of earthquake catalogues [JGR  Space Physics 119,  9192–9206  (2014) https://doi.org/10.1002/2014JA020580 ] was proposed in 2015 [PNAS 112, 986-989 (2015) https://doi.org/10.1073/pnas.1422893112 ] to estimate the epicentral location of major earthquakes by studying the fluctuations of the spatiotemporal variations of the variability [EPL 91, 59001, 2010 https://doi.org/10.1209/0295-5075/91/59001 ] of the order parameter of seismicity introduced in natural time analysis. The results show that by dividing the Japanese region, for example, into small areas and carrying out the variability calculation on them one finds that some small areas exhibit a minimum of the variability almost simultaneously with the large area and such small areas clustered within a few hundred kilometers from the actual epicenter of the related mainshocks.” 

Moreover, the following typos should be corrected:

l.39 “this work,”, l.46 “the Frenkel’s model”, l.64 rephrase “kernel for the”, l.84 “took as threshold M_w=5.5”, ll.114, 122 do not indent, l.124 “that different kernel”, l.140 “indeed it leads”, l.146 “seismic activity”, l.150 “cells”, l.162 do not indent, l.235 “Adopted” -> “Adaptive”, l.262 “5. The”, l.270 “Discussion”, and l.292 “straightforward than that of Wang”.

In view of the above, I strongly advise the authors to revise their ms along the lines of the points mentioned above. I will be glad to suggest publication of an appropriately revised ms.

Author Response

REVIEWER #3

This manuscript (ms) by Taroni and Akinci presents a new smoothed seismicity approach to include aftershocks and foreshocks in spatial earthquake forecasting. The application in made to the Global Mw≥5.5 seismicity and specific examples are discussed in detail for large regions around Indonesia, Mexico and Chile. The authors suggest an original and interesting improvement and the results convince the reader for the significance of the suggested new approach. The ms, however, although well written, needs improvements in the presentation. These improvements are:

COMMENT#1: In the Introduction, there is no comment on the very important and modern contribution of Earthquake Nowcasting which is closely related to the subject under study.

ANSWER#1:

We agree with the reviewer, earthquake nowcasting is an important modern contribution to the earthquake forecasting problem, but is not completely related to the PSHA approach (nowcasting is more focused on the short and medium-term forecast, PSHA is focused on the long-term forecast). We added a statement in the “Discussion” section, that also takes into account the Comment#5 of the reviewer. See the Answer#5 for more details.

COMMENT#2: The letter size in all figures should be increased because it is very hard to read the labels. Moreover, in the multipaneled figures 2, 3, 4 and 5 panel labels (a), (b), (c) should be added.

ANSWER#2:

We accepted and modified all requested variations. We underline that the final dimension of the figures will depend on the final proof of this manuscript, and will be chosen to make all the labels easily readable.

COMMENT#3: The acronym NN seems to be multiply defined in the lines 130 and 183. This should be avoided. Moreover, for the readers’ better understanding an example of smoothing distances with actual values should be given either in the lines 122-134 or in the lines 181-184.  

ANSWER#3:

We removed the multiple NN definition and give NN values in section 4 ”Likelihood Testing for Spatial Variation of Seismicity”. Finally, the nearest neighbor numbers and the correlation distances are calculated through maximum-likelihood optimization for the four smoothed seismicity models using the last ten years of the learning catalogue (2000-2009). The results of these estimations are summarized and provided in Table 2.

COMMENT#4: The discussion in the lines 235 to 245 mixes the notion of “the smallest log-likelihood values” with the “better performance in forecasting future earthquakes” (simply -29632 > -31297). The authors should rephrase this paragraph to avoid confusion in the readers.

ANSWER#4:

Thanks for the suggestion; we added a statement to clarify our thoughts, and we modified the text according to the rule “larger LL → nearest to zero”:

“For a correct interpretation of the models’ LL, we recall that: large LL values (i.e. the ones nearest to zero) indicate relatively good performances of the models, small LL values (i.e. the ones further from zero) indicate relative bad performances of the models. In general, our results show that the adaptive smoothed seismicity models (Adoptive and Adaptive_corrected) produce the larger LL values and reveal better forecasting performance with respect to those from the fixed smoothed ones (Fixed and Fixed_corrected). The LL values are -29632 and -29639 for the corrected and uncorrected adaptive smoothed models, while it is -31297 and -31198 in the case of the fixed corrected and corrected smoothed seismicity models, respectively (Table 3). The largest LL value calculated for the adopted smoothed seismicity models arises from the use of the correction parameter including the foreshocks and aftershocks in the global catalog.”

COMMENT#5: Last but not least, the present ms suffers from the following very serious shortcoming, which should be necessarily removed: Although the authors state in their abstract (line 12) that here they “present a technique to better determine future earthquakes in space”, surprisingly do not mentioned at all in their ms the recent work highlighted by PNAS, entitled “Spatiotemporal variations of seismicity before major earthquakes in the Japanese area and their relation with the epicentral locations” [PNAS 112, 986-989 (2015)], which presents a modern approach to estimate the epicentral location of forthcoming earthquakes. Thus, I would suggest to the authors to add the following (probably as a separate paragraph in their “Discussion” Section 6):“A different approach based on natural time analysis of earthquake catalogues [JGR  Space Physics 119,  9192–9206  (2014) https://doi.org/10.1002/2014JA020580 ] was proposed in 2015 [PNAS 112, 986-989 (2015) https://doi.org/10.1073/pnas.1422893112 ] to estimate the epicentral location of major earthquakes by studying the fluctuations of the spatiotemporal variations of the variability [EPL 91, 59001, 2010 https://doi.org/10.1209/0295-5075/91/59001 ] of the order parameter of seismicity introduced in natural time analysis. The results show that by dividing the Japanese region, for example, into small areas and carrying out the variability calculation on them one finds that some small areas exhibit a minimum of the variability almost simultaneously with the large area and such small areas clustered within a few hundred kilometers from the actual epicenter of the related mainshocks.” 

ANSWER#5:

We thank to referee for informing us about these studies, which are now mentioned and cited in the new version of the manuscript, using two simple statements (to avoid burdening of the text). These statements also include the reply to Comment#1 on Earthquake Nowcasting (we cited the Rundle et al. 2016 paper):

 “Our method is based on the assumption of stationarity of the seismicity (usually accepted in long-term modeling); however, working in smaller time and spatial scales, some regions may exhibit different spatiotemporal variations, useful to forecast stronger seismic events (Sarlis e al., 2010 and 2015). Abandoning the stationarity assumption, smaller earthquakes can also be used to try to determine the current state of the seismic cycle (Rundle et al. 2016), and then identify possible temporal variations in the long-term seismic rates.”

Moreover, the following typos should be corrected:

COMMENT#6: l.39 “this work,”, l.46 “the Frenkel’s model”, l.64 rephrase “kernel for the”, l.84 “took as threshold M_w=5.5”, ll.114, 122 do not indent, l.124 “that different kernel”, l.140 “indeed it leads”, l.146 “seismic activity”, l.150 “cells”, l.162 do not indent, l.235 “Adopted” -> “Adaptive”, l.262 “5. The”, l.270 “Discussion”, and l.292 “straightforward than that of Wang”.

ANSWER#6:

We have corrected all those suggested typos, thanks.

In view of the above, I strongly advise the authors to revise their ms along the lines of the points mentioned above. I will be glad to suggest publication of an appropriately revised ms.

We thank the reviewer for the positive answer!

Reviewer 4 Report

This study aimed to improve the seismicity model used in the context of Probabilistic Seismic Hazard Assessment (PSHA). The authors considered a global seismicity catalogue and evaluate the gain in spatial information obtained by using different techniques for seismicity model smoothing. More specifically, they propose a correction that allows inclusion of foreshocks and aftershocks with a simple equal weight given to all events of a sequence. Their results corroborate previous studies regarding the adaptive smoothing technique being superior to the fixed smoothing technique. Their correction that includes foreshocks/aftershocks also shows an improvement in log-likelihood minimization for a global test catalogue compared to the exclusion of these events. This study show a promising improvement to spatial estimation of seismic hazard and proposes an easier to implement method for the inclusion of foreshocks/aftershocks than previously proposed by other authors (Wang et al., 2011).

For the reader unfamiliar with the authors’ previous studies and the problem of catalogue declustering in the context of PHSA, it was difficult to grasp the significance of this study in the context of seismic hazard assessment after a first reading of the manuscript. For example, the authors clearly stated the need for declustering catalogues for the purpose of avoiding bias in PSHA due to the inter-event dependence, yet this study includes foreshocks/aftershocks and it is confusing how this does not violate the requirements that are needed for the next steps of PSHA analysis. Added details and clarifications in the introduction and discussion as suggested below would help better understand the novelty of the study and its impact on PSHA and make this paper more impactful in a journal like Applied Sciences that caters to readers of very varied disciplines. The results also need to be better supported by adding MLE results for the range of parameters tried instead of only the maximum values. I consider these needed changes to be a major revision, hence I recommend the manuscript to be published after major revision.

Major comments:

  • Authors need to clearly explain/demonstrate how including aftershocks/foreshocks is not violating the required PHSA assumption of independent events as this is confusing for the reader. Can the author’s proposed improvements be used in the context of PSHA?
  • The inclusion of foreshocks/aftershocks is not new as the authors mention in paragraph 56-69. The innovation from this study, which is giving an equal normalized weight to all events of a sequence is claimed to be superior to Wang et al. (2011), who use a variable weight for each event of a sequence, giving more weight to the mainshock in terms of performance. It would have been more appropriate for this study to compare their results to the Wang (2011) approach in terms of performance, not just simplicity or ease of implementation. Instead the authors compare their result to the complete exclusion of foreshock/aftershock, which does not allow the readers to understand how this approach improves upon Wang et al. (2011). I suggest the reader compare their correction for fore/aftershocks to the one of Wang 2011 in terms of performance.
  • It is unclear for a reader not already familiar with the topic of declustering and smoothing of seismicity catalogues for the purpose of hazard analysis how this study is different from Taroni and Akinci 2021, which also found that removing aftershocks/foreshocks during the declustering procedure underestimates seismicity rates and seismic hazard. The introduction should better introduce the different clustering techniques and how this study distinguishes itself from Taroni and Akinci 2021.
  • The optimal NN parameter being the lower bound of 1 in the MLE seems suspect. Instead of Table 2, please provide two plots showing MLE as a function sigma and NN respectively, with a separate curve for the corrected and uncorrected case. Indicate on the plot the maximum values. 
  • In the MLE, explain how considering a specific tectonic region as in figure 2 as opposed to a global catalogue would affect your results; what would be the optimum values for the three regions in figure 2? Can the estimated values be biased by for example volcanic event catalogues that are not appropriately treated by declustering? Please discuss in the manuscript the potential biases caused by considering a global catalogue as opposed to a regional one.

Minor issues/comments:

  • Abstract should be one paragraph (remove break at line 27-28). 
  • Line 39: Please explain in the introduction the goal of these two approaches in the context of PHSA.
  • Line 88: For reproducibility, the parameters used for declustering with ZMAP should be provided.
  • Line 112: Provide references for the equation if derived from previous studies.
  • Line 127-130: Give references for example of fixed smoothing and adaptive smoothing model approaches..
  • Line 151: Please briefly summarize from Taroni and Akinci 2021 how these two factors impact your results.
  • Line 161: I suggest to use one letter for Log-Likelihood, e.g. ℒ or ℓ or simply L, as this is more common.
  • Line 169: “in case we assume the independence between the observations in the set X”. Since you are including foreshocks and aftershocks in your catalogue, you are including observations dependent on each other, so it seems this assumption would not hold. Please clarify.

Issues with figures:

  • Figure 1: Add blue boxes to show zoom-in areas instead of only letters to get a sense of relative scale.
  • Figures 1 and 2: To make it clearer what is land or water in those maps, please put a fill color, for example a gray shade for the polygons representing continents or some shade of blue for oceans (in that latter case change color of blue seismicity points). Consider adding shaded terrain relief to add info on the regional geography.
  • Figures 1 to 5: Write “latitude” and “longitude” fully and increase label size
  • Figures 3 to 5: please add colorbar labels and units and add labels a), b), etc. for each subplot.
  • Please combine figures 3 and 4 into one figure of 4 subplots for better side by side comparison and use the same colobar for all 4 subplots.
  • Figure 5: please add titles to the subplots giving the name of the region
  • Figure 5: It is difficult with this colormap to see the threshold for positive and negative differences. Instead, a more appropriate colormap would be blue-white-red, with white being the zero value, and red and blue being positive and negative respectively.

The attached PDF includes corrections for typos/English styling .

Author Response

See attached PDF

Round 2

Reviewer 4 Report

I thank the authors for carefully considering the previous review report. They made appropriate changes to the figures and text that improve clarity of the manuscript and acknowledge limitations of the proposed technique. There are some formatting and spelling mistakes lurking in some places (see some examples below) that can be taken care of my the copyediting team. I can now recommend publication by Applied Sciences of this paper.

Minor revisions:

  • L70: "delcustering" --> declustering
  • L176: Missing new line before header "4. Likelihood ..."
  • Figure 3: Panel labels should be "a)", "b)", etc. instead of "(a", "(b"
  • In the caption: (L221) "Blu" --> Blue
  • L407: Missing new line before new citation
  • L388 and L407: Format of citation differs from others.